

# Satellite-detected sea surface chlorophyll-a blooms in the Japan/East Sea: magnitude and timing

Dingqi Wang[1,2], Guohong Fang[1,3,4], Shumin Jiang[1,2], Qinzeng Xu[1], Guanlin Wang[1,3,4], Zexun Wei[1,3,4], Yonggang Wang[1,3,4], and Tengfei Xu[1,3,4]

[1] First Institute of Oceanography and Key Laboratory of Marine Science and Numerical Modeling, Ministry of Natural Resources, Qingdao 266061, China
[2] College of Oceanic and Atmospheric Sciences, Ocean University of China, Qingdao, 266061, China
[3] Laboratory for Regional Oceanography and Numerical Modeling, Pilot National Laboratory for Marine Science and Technology, Qingdao 266237, China
[4] Shandong Key Laboratory of Marine Science and Numerical Modeling, Qingdao 266061, China

*Correspondence to*: Tengfei Xu (xutengfei@fio.org.cn)

**Abstract.** The Japan/East Sea (JES) is known as a mid-latitude "Miniature Ocean" that features multiscale oceanic dynamics processes. We investigate the variability of the sea surface chlorophyll-a concentration (SSC) and bloom timing in the JES based on satellite remote sensing products spanning 1998–2019. The JES SSC exhibits strong seasonal variability and blooms twice annually, which are mainly governed by the physical environmental conditions. However, the influences of local oceanic dynamic processes (e.g., upwelling, oceanic fronts, mesoscale eddies, and near-inertial oscillations) on the bloom magnitude and timing of the entire JES are not critical, compared with the PAT and stratification. In addition, significant interannual variabilities of spring bloom magnitude occur along the JES's northwestern coast, and that of fall bloom magnitude occur in the deep Japan Basin. For spring bloom, the interannual variability of the bloom timing (initiation timing, termination timing and duration), which significantly affect the interannual bloom magnitude anomalies, are correlated with climate modes such as AO and ENSO. For fall bloom, on the interannual time scale, the bloom duration is mainly affected by the initiation timing. Both of them have a significant influence on the bloom magnitude. The initiation/termination timing of spring blooms has shifted earlier by 0.37/0.45 days annually along the JES's northwestern coast; the counterpart of fall blooms has shifted 0.49/1.28 days earlier annually in the deep Japan Basin.

**Keywords:** sea surface chlorophyll-a concentration (SSC); Japan/East Sea (JES); spring bloom; fall bloom; interannual variability

## 1 Introduction

As the most common primary producer in the marine food chain, phytoplankton respond quickly to changes in their physical environment and are thus sensitive to climate change (Hays et al., 2005). Satellite-based ocean color observations can provide sea surface chlorophyll-a concentrations (SSCs) over the global oceans, which are commonly used for estimating phytoplankton concentrations (Banse, 1977; Liu and Wang, 2022; Taboada et al., 2019; Wang et al., 2021). The Japan/East Sea (JES) is a semi-enclosed marginal sea located in the middle latitudes of the Northwest Pacific Ocean (Fig. 1). The JES involves multi-scale oceanic dynamic processes (Ichiye, 1984), thereby resulting in complicate SSC variations by changing nutrient supply (Park et al., 2020). For instance, the



warm Tsushima Current and cold Liman Current forms a thermal boundary in Sea Surface Temperature (SST) in the
38~40°N region, namely, a subpolar front (Yamada et al., 2004). The subpolar front induces nutrients accumulation
and further favor SSC increases (Lee et al., 2009). Featuring strong offshore monsoons, the JES is abundant with
wind-driven Ekman upwelling; this upwelling can carry nutrient-rich water from deeper layers and thus nourish
phytoplankton, resulting in heightened SSC (Park et al. 2020). Sea ice-melted water carries high nutrients,
promoting the increase of SSCs along the coast of Russia (Martin and Kawase 1998; Nihashi et al., 2017; Park et al.
2014). Mesoscale eddies and typhoon also play important roles in SSC variations (Ji et al., 2021; Liu et al., 2019;
Maúre et al., 2017). Overall, the upper dynamics in the JES are quite complex, and different processes may
counteract with each other in different periods, which leads to the complexities of SSC variations.
The phytoplankton concentrations in the JES blooms twice each year, i.e., the spring bloom (March–May) and the
fall bloom (October–November), both of which can be detected by satellite observed SSCs (Ishizaka and Yamada,
2019; Jo et al., 2014; Kim et al., 2000). Spring blooms, occurring in almost the entire JES, are initiated earlier in the
southern region and later in the northern region (Maúre et al., 2017; Yamada et al., 2004). Basically, spring blooms
can be explained by the critical depth hypothesis (Sverdrup, 1953). In winter, phytoplankton growth is light-limited
(due to deep mixing) rather than being nutrient limited. In spring, the mixed layer becomes shallower than the
critical depth, resulting in unlimited light availability to initiate spring bloom (Kim et al., 2000). The critical depth
hypothesis is extended by considering the relaxation of turbulent mixing conditions associated with surface cooling
(Taylor & Ferrari, 2011) and weakened wind stress (Kim et al., 2007; Lee et al., 2015). Additionally, eddy-driven
stratification could regulate the initiation timing of the spring bloom (Mahadevan et al., 2012), with
anticyclonic/cyclonic eddy playing different mechanisms, respectively (Maúre et al., 2017). In comparison, fall
blooms are much weaker and occur only in the western JES (Kim et al., 2007; Yamada et al., 2004). The fall bloom
relies on upward transport of nutrient-rich waters from the deeper layers. It is expected to start when the MLD
deepens and becomes equal to the critical depth, as driven by enhanced wind and surface cooling that favor
destratification of water column (Kim et al., 2000; Yamada et al., 2004). However, since there are different
dynamics and corresponded physical environment factors during spring and fall blooms, it is still a matter of debate
about the dominant factors that favor and/or restrict SSC during their bloom and decline stages (Maúre et al., 2017).
Additionally, Park et al. (2022) proposed that the interannual variability of SSC is significant in the JES, because
the contribution of seasonal cycles to the total variance in SSC variability is less than 30%. Park et al. (2020)
suggested that interannual SSC anomalies have only one dominant annual peak that occurs in March or April. The
interannual SSC anomaly along the JES's northwestern coast in spring is highly related to the sea ice concentration
(SIC) in the Tartar Strait in the previous winter from 1999 to 2007, as more winter SIC would provide more
nutrients when melting (Park et al., 2014). For the initial timing, the spring blooms tend to start early/late in El
Niño/La Niña years in response to weak/strong wind speed-induced turbulent mixing (Yamada et al., 2004).
Meanwhile, the El Niño-Southern Oscillation (ENSO) events can influence the strength and direction of the
Tsushima Warm Current to modify the location and maintenance of the subpolar front, which in turn influences the
initiation region and timing of the spring bloom (Yoo and Kim,2004). In comparison, the interannual variability of
fall blooms is much weaker, and their initiation timing is less correlated to ENSO-induced wind speed anomalies



76 over the JES (Yamada et al., 2004). During positive Arctic Oscillation (AO), the SSC anomaly in spring might

77 slightly increase due to the weakening of wind speed and the weak increase of SST in previous winter (Park et al.,

78 2022). However, it is not clear that whether the interannual variability of JES SSC related to climate modes, as the

79 previous research results are based on composite analyses without a test of confidence level.

80  Furthermore, the temporal variation in bloom magnitude and timing, including initiation timing, termination

81 timing and duration, has significant ecological and biogeochemical influences (Behrenfeld and Boss, 2017). The

82 strong phytoplankton blooms will lead to the imbalance of marine ecosystem, resulting in huge economic losses,

83 especially harmful algal blooms (Ok et al., 2021). Based on observation data from 1972 to 2002, Yamada and

84 Ishizaka (2006) proposed that spring bloom in the southern JES started relatively earlier in mid-1980s, resulting in a

85 regime shift of the community structure of spring diatom from cold water species to warm water species, which are

86 small and adapted to oligotrophic condition. Additionally, in the southern JES, the recruitment of Japanese sardine

87 was positively affected by delays in the start and end timing of the spring bloom, because the overlap of bloom

88 duration and sardine larval periods prolonged (Kodama et al., 2018). However, there is a lack of a study

89 synthesizing variability of bloom magnitude and timing in the seasonal and interannual cycle in the whole JES area.

90 In this study, we attempt to reveal the favorable/restricting factors during the SSC raise and decline stages, and

91 investigate the variations in bloom magnitude and timing in spring and fall, as well as their related physical

92 environmental factors and climate modes, including ENSO and AO.

93 **2 Data and Methods**

94 **2.1 Data**

95 The daily SSC data is a Level-4 product providing globally cloud-free estimations during 1998–2019 at a 4-km

96 resolution. The product is published by the Copernicus Marine Environment Monitoring Service (CMEMS), and has

97 merged ocean color observations from multiple sourced satellites (Garnesson et al., 2019). The Level-4 product

98 preserves the information of Level-3 product, and can resolve the SSC variations with time scales longer than

99 intraseasonal (Garnesson et al., 2019; Xu et al., 2021). The *in-situ* chlorophyll-a concentration measurements at 10

100 m depth obtained from the World Ocean Database 2018 (WOD18), are used to validate the satellite-derived SSC

101 data in the JES (Boyer et al., 2018). A total of 1172 chlorophyll-a profiles during 1998–2019 was obtained (Fig. 1).

102 The satellite-derived SSC data are generally consistent with the *in-situ* observations, with a high correlation

103 coefficient of 0.79 ($p<0.01$) (Fig. 2). Thus, the satellite-detected SSC data used in this study is reliable for the

104 following research.

105  The photosynthetically active radiation (PAR) and its attenuation coefficient ($k$) are provided by the European

106 Service for Ocean Colour with a 4-km horizontal resolution (Maritorena et al., 2010). Satellite-based SIC data in the

107 Tatar Strait (47°–52°N, 139°–142°E) are obtained from the National Snow and Ice Data Center of the National

108 Oceanic and Atmospheric Administration (Meier et al., 2011). The sea surface height (SSH) and sea surface

109 geostrophic current anomalies, with a horizontal resolution of 0.25°×0.25°, are derived from the daily gridded

110 absolute dynamic topography products version 5 and are distributed by the Archiving, Validation, and Interpretation



of Satellite Oceanography (Ducet et al., 2000). Daily SST are derived from the CMEMS with a 0.05°×0.05°
horizontal resolution (Good et al., 2020). The surface wind vector data are provided by the European Centre for
Medium-Range Weather Forecasts ERA5 high-resolution reanalysis project, with a horizontal resolution of
0.25°×0.25° (Hersbach and Dick, 2016). The monthly Niño3.4 and Arctic Oscillation (AO) indices data are
collected from the Koninklijk Nederlands Meteorologisch Instituut (KNMI) climate explorer (Trouet et al., 2013). In
this study, all data described above cover the period from 1 January 1998 to 31 December 2019. Climatological
monthly mean temperature, salinity and nutrient profiles are obtained from the World Ocean Atlas 2018 (WOA18)
(Garcia et al., 2019). Details for the datasets used in this study are presented in Table 1.

**Table 1. Parameters of the datasets used in this study.**

| Source | Variables | Temporal Coverage | Temporal resolution | Spatial resolution | URL |
|---|---|---|---|---|---|
| CMEMS | SSC | 1998–2019 | daily | 4-km | https://resources.marine.copernicus.eu |
| WOD18 | SSC | 1998–2019 | - | - | https://www.ncei.noaa.gov/products/world-ocean-database |
| GlobColour | PAR and $k$ | 1998–2019 | daily | 0.25°×0.25° | http://www.globcolour.info/ |
| NSIDC | SIC | 1998–2019 | daily | - | http://nsidc.org/ |
| AVISO | SSH, and sea surface geostrophic current anomalies | 1998–2019 | daily | 0.25°×0.25° | http://www.aviso.altimetry.fr/duacs/ |
| CMEMS | SST | 1998–2019 | daily | 0.05°×0.05° | https://resources.marine.copernicus.eu |
| ERA5 | surface wind | 1998–2019 | 6-hourly | 0.25°×0.25° | https://cds.climate.copernicus.eu/cdsapp#!/dataset/reanalysis-era5-single-levels?tab=form |
| KNMI | Niño3.4, and AO indices | 1998–2019 | monthly | - | http://climexp.knmi.nl/selectindex.cgi?id=someone@somewhere |
| WOA18 | temperature, salinity and nutrient profiles | - | monthly | 0.25°×0.25° | https://www.ncei.noaa.gov/access/world-ocean-atlas-2018 |


**2.2 Methods**
A phytoplankton bloom is defined as a period while the SSC exceeds a certain percentage (20%) of its annual
median value over a duration longer than three weeks (spring bloom) or one week (fall bloom), consistent with that
used in Maúre et al. (2017). As shown in Fig. 3, each grid point has its own threshold SSC value for a bloom. To
identify blooms for the entire JES, the threshold SSC value calculated from the area-averaged SSC data is about
0.55 mg m$^{-3}$.
The Ekman pumping velocity is calculated as follows:





$$w_E = - \nabla \times (\frac{\boldsymbol{\tau}}{\rho_0 f})$$

where $\rho_0$=1.025 × 10³ kg/m³, is the mean sea water density, $f$ is the Coriolis parameter, and $\tau$ is the wind stress
derived from ERA5 wind field product.
The eddy kinetic energy (EKE) is calculated as follows

$$EKE = \frac{1}{2}(u'^2 + v'^2)$$

where $u'$ and $v'$ are the sea surface geostrophic current anomalies.
The wind-induced near-inertial energy flux (WNEF) is estimated by a simple slab mixed layer model as follows
(Pollard & Millard, 1970):

$$\Pi(H) = Re(\boldsymbol{Z} \cdot \boldsymbol{\tau}^*)$$

where $\boldsymbol{Z}$=$u+iv$ represents the mixed-layer current, $H$ is the area-averaged climatological monthly mean MLD, and $\tau^*$
is the conjugate of $\boldsymbol{\tau}$. A spectral solution through a two-sided Fourier transform is used here, expressed as the
following equation:

$$\widehat{\boldsymbol{Z}}(\sigma) = \frac{\widehat{\boldsymbol{\tau}}(\sigma)}{H} \frac{r - i(f + \sigma)}{r^2 - i(f + \sigma)^2}$$

where $r(\sigma) = r_0(1 - e^{-\sigma^2/2\sigma_c^2})$ is the frequency-dependent damping parameter, $\sigma$ represents the angular frequency,
$r_0$=0.15$f$ and $\sigma_c$=$f$/2 (Alford, 2003).
The Brunt Väisälä frequency, $N$, is used to estimate the vertical stability in the upper 200 m of the ocean as
follows:

$$N = \sqrt{-\frac{g}{\rho}(\frac{d\rho}{dz})}$$

where $g$ is the gravitational acceleration, $\rho$ is the potential density of sea water, and $z$ is the depth.
The MLD is calculated by defining a temperature threshold of 0.3°C from 10 m, as suggested by Jo et al. (2014).
Both $N$ and MLD are calculated from the WOA18 data.
The gradient-based edge detection algorithm is employed to detect SST fronts (Castelao & Wang, 2014). The
monthly frontal probability (FP), which represents the occurrence frequency of SST fronts, is defined as the ratio
between the occurrence days of frontal conditions and the total days in the corresponding month.
The critical depth (CRD) is computed as follows:

$$CRD = \frac{I_0}{kI_c}$$

where $I_0$ is the PAR (E m⁻² d⁻¹) and $k$ denotes the attenuation coefficient of PAR. The compensation light intensity $I_c$
is taken as 3.8 E m⁻² d⁻¹ (Kim et al., 2000).
Principal component analysis (PCA) is a simple, non-parametric method for extracting relevant information from
confusing data sets. PCA reduces such datasets to a lower dimension to increase interpretability, but simultaneously
preserves as much 'variability' (i.e. statistical information) as possible, based on the principle of variance
maximization (Jolliffe and Cadima, 2016; Shlens, 2014; Trombetta et al., 2019). Finding new uncorrelated variables,
the principal components, reduces to solving an eigenvalue/eigenvector problem, and the new variables are defined





by the dataset at hand, not *a priori*, hence making PCA an adaptive data analysis technique (Jolliffe & Cadima,
2016). This technique is popular for analysis of atmospheric and oceanic data, and is often referred to as Empirical
orthogonal function (EOF) called by Lorenz (1956). Both names are commonly used, and the essence of the two is
the same. However, EOF often examines the variability in data through space, which is widely used to extract
patterns, while PCA is often used to highlight the relationships between different variables over time (Hannachi et
al., 2007; Trombetta et al., 2019). In this study, PCA is used to identify the relevant factors that might influence the
seasonal SSC cycle following Trombetta et al. (2019), while EOF is employed to explain the spatiotemporal
distribution of SSC in the JES on seasonal and interannual time scales (Greene et al., 2019). Prior to the PCA and
EOF analysis, the monthly mean SSC data is logarithmically transformed due to its lognormal distribution
(Campbell, 1995), as suggested in previous studies (Dandonneau, 1992; Xu et al., 2021).

## 3 Results

### 3.1 Seasonal variability of bloom magnitude

The seasonal cycle of the SSC in the JES shows double peaks associated with the bimodal blooms of phytoplankton
concentrations in spring and fall, respectively (Fig. 4). The JES SSC is at a low level in boreal winter, with values
generally smaller than the threshold SSC value (0.55 mg m$^{-3}$) used to identify blooms for the entire JES (Fig. 4a–4b).
In spring, a bloom occurs around the subpolar front region in March and then extends southward and northward to
cover most areas of the JES in April, with SSC values up to 10 mg m$^{-3}$ (Fig. 4c and 4d). In May, SSC begins to
decrease from the southeastern JES (Fig. 4e). From June to September, the SSC values are smaller than 0.2 mg m$^{-3}$
in most regions of the JES except for the Tartar Strait (Fig. 4f–4i). Fall bloom begins in October, when the SSCs
increase slightly over the entire JES, with smaller magnitudes than spring bloom (Fig. 4j). The SSCs are higher
along the Russian and Korean coasts until December (Fig. 4k and 4l).

The seasonal evolution of SSC can be derived by applying EOF analysis on the logarithmic monthly mean SSC

(Fig. 5). The first three leading EOF modes explain 64.2%, 8.8% and 3.6% of the total variance. The first EOF mode
(EOF1) essentially represents the basin-scale seasonal variability in the JES SSC (Fig. 5). The EOF1 is reminiscent
of the SSC distribution in April, showing positive values in the entire JES with relatively small magnitudes in the
Japan Basin (Fig. 5a and Fig. 5d). The seasonal time coefficient of EOF1 coincides with the area-averaged SSC in
the JES, showing double peaks of 1.36±0.28 and 0.74±0.07 mg m$^{-3}$ in April and November, respectively (Fig. 5b).
During the December–February and June–September periods, the area-averaged SSC are smaller than the threshold
criterion, suggesting poor primary production during these stages.

Previous investigations have qualitatively discussed the potential factors accounting for the seasonal raise and

decline in SSC. These factors are summarized as follows (Fig. 6).

(1) PAR (Fig. 6a). In boreal winter, a lower solar altitude angle results in less PAR with an average value of 30.55

E m$^{-2}$ d$^{-1}$. In spring or summer, shortwave radiation increases due to the raise of solar altitude angle.

(2) Nutrient supply (Fig. 6b and 6c). The upper layers of the JES are rich in nutrients during the boreal winter.

Both nitrogen and phosphate are rapidly consumed during spring blooms, with the nitrogen concentration decreasing



from a peak value of 7.68 in March to 1.06 μmol kg⁻¹ in May and the phosphate concentration decreasing from a
peak value of 0.58 in March to 0.25 μmol kg⁻¹ in May. The nutrients supplementation is poor from June to
September when SSC is at a low level, and enhance following October.
(3) The MLD and CRD (Fig. 6d). Due to the annual cycle of PAR, the CRD is deepening from January and
becomes equal to the shoaling MLD around February, when SSC becomes to increase rapidly. In autumn, SSC
reaches the second peak around November when the deepening MLD the shoaling CRD coincides. However, the
initiation timing of spring bloom and fall bloom are later than February and earlier than November, respectively.
This time bias might be caused by the value of the $I_c$, as the temporal variations of $I_c$, of 6.3 E m⁻² d⁻¹ during spring
and 1.4 E m⁻² d⁻¹ during fall, is not considered in the calculation of CRD (Kim et al., 2000). Thus, the Sverdrup
hypothesis (Sverdrup, 1953) is basically applicable to explain the bloom initiation in the JES, indicating that the
variations in the JES SSCs are mainly governed by the physical environmental conditions.
(4) The JES shows enhanced stratification from March to August (Fig. 6e) due to weakened wind speeds (Fig. 6f)
and strengthened buoyancy fluxes contributed by both surface warming (Fig. 6g) and sea ice melting (Fig. 6h). In
contrast, destratification occurs from September until the following February. Stratification is basically out of phase
with the MLD; this process can be explained by the fact that strong/weak stratification favors MLD
shoaling/deepening. It is worth noting that sea ice melting is also an important nutrient supply source in boreal
spring (Park et al., 2014). Since the temporal variations of BV frequency and SST are almost identical, SST is used
as the index to quantify stratification in the following analysis.
(5) Ocean dynamics contribute to vertical nutrient-water transport (Fig. 6i-6l). Ocean dynamics dominate the
upper layer nutrients only in boreal summer and fall, when the upper JES is oligotrophic. After October, nutrients
increase in accordance with the enhanced upwelling and frontal probability (Fig. 6i and 6j), and under these
processes, deeper nutrient-rich waters are entrained to the upper layer. The EKE shows larger energy in the southern
JES from August through December (Fig. 6k). The WNEF shows larger energy in response to the more frequent
typhoons in boreal summer and fall (Fig. 6l). However, since mesoscale eddies and typhoon-induced near-inertial
oscillations generally occur in the southeastern JES, an area that also has strong stratification, it is suggested that
only a few typhoons appreciably increase the nutrient supplies in this area (Iwasaki, 2020). Moreover, the high
nutrients induced by WNEF are locally distributed along typhoon tracks and thus do not significantly contribute to
the area-averaged nutrients over the entire JES. As a result, the monthly averaged nutrients are not efficiently
entrained from the deeper layer to the upper layer to nourish the phytoplankton during the June-September period.
Nevertheless, the EKE and WNEF remain at relatively high levels until November; therefore, these factors may still
contribute to vertical nutrient transport, albeit they are not critical factors.
To identify the dominant and limiting factors affecting the seasonal variability of SSC in the JES, we classify the
JES SSC evolution into four stages: the raise stage (January–April) and decline stage (April–July) of spring blooms,
and the raise stage (July–November) and decline stage (November–next January) of fall blooms. The relationships
between SSC and environmental factors during different stages are examined by PCA analysis (Fig. 7). During the
raise stage of spring bloom, the SSC is positively correlated with PAR and negatively correlated with wind speed
and SIC, suggesting dominant factors inducing spring blooms include increased PAR, weakened winds and sea ice





melting. In comparison, oceanic dynamics are not correlated with SSC, as revealed by that they are approximately
orthogonal to each other. During the decline stage of spring bloom, the stratification shows a symmetrically opposite
relationship with SSC, suggesting that SSC decline is related to enhanced stratification. The PAR is saturated and is
thereby orthogonal to SSC, and oceanic dynamics are depressed at this time and thus rarely impact SSC. The
coincident directions of SSC and FP occur because subpolar fronts are also undergoing a weakening phase at this
time (Park et al., 2007). During the raise stage of fall blooms, SSC is positively correlated with the wind speed,
upwelling and oceanic fronts; and negatively correlated with PAR and stratification. These results suggest that
destratification, strengthening winds and associated upwelling, and enhanced oceanic fronts, favors an increase in
SSC by entraining more nutrient-rich waters to the upper layer. The declining PAR tends to play a negative role and
acts as a limiting factor for phytoplankton growth when it is reduced to a certain extent. During the decline stage of
fall blooms, the winds and associated upwelling tend to enhance the upward transport of nutrient-rich waters to the
upper JES. However, limited by decreased PAR and increased sea ice, SSC is still suppressed, resulting in the decay
of fall blooms. It is worth noting that the nutrients accumulated during the decline stage of a fall bloom essentially
contribute to the following spring bloom.

**3.2 Interannual variability of bloom magnitude**

The spring and fall blooms are attributed to different mechanisms. Therefore, we conduct EOF analyses for the
interannual SSC anomalies (remove seasonal variability) during spring (March–May) and fall (October–November),
respectively. EOF1 accounts for 21.2% and 27.0% of the total variances for the spring and fall blooms, respectively.
The significant interannual variability of SSC in spring and fall occurs along the JES's northwestern coast (Fig. 8a),
and in the deep Japan Basin (Fig. 8b), respectively . The positive spring SSC events occur in 2001, 2002, 2009 and
2016, while negative spring SSC events occur in 2005, 2006, 2012 and 2013 (Fig. 8c). Positive fall SSC events
occur in 1999 and 2015, while negative ones occur in 2008 and 2012 (Fig. 8d). Additionally, there is decadal
variability in fall blooms, with positive phases during 1998–2002, and 2014–2019, whereas negative phase during
2003–2013.
The the interannual variability of JES SSC is positively correlated with PAR, with correlation coefficient ($R$) of
0.37 at a confidence level of $p < 0.1$ in spring and of $R = 0.12$, $p > 0.1$ in fall (Fig. 9a and 9c). For stratification
anomalies, the correlations are $R = 0.55$, $p < 0.01$ in spring and $R = –0.44$, $p < 0.05$ in fall (Fig. 9b and 9d). These
correlations suggest that photosynthetic activity positively contributes to the interannual SSC variability in the JES
in spring. Meanwhile, strong stratification favors positive SSC anomalies in spring, as explained by the critical depth
hypothesis. In contrast, PAR anomalies are not significantly correlated with SSC anomalies in fall, whereas
stratification shows a significant negative correlation, suggesting that stronger stratification leads to smaller SSCs by
inhibiting the upward transport of nutrient-rich waters to the mixed layer.
By comparing Fig. 10a and 10b, the interannual variability of JES SSC is more statistically correlated with AO ($R$
$= 0.38$, $p <0.1$) than ENSO ($R = 0.26$, $p > 0.1$) in spring. This correlations suggest that the spring bloom magnitude
would increase during positive AO, which demonstrates the previous research results (Park et al., 2022). However,





the climate modes have little effect on modulating the interannual variability of fall bloom magnitude, with low
correlation coefficients that far lower than the 90% confidence level, as shown in Fig. 10c and 10d.

**3.3 Bloom timing**

Spring bloom initiates in the southern and southeastern JES in February–March and in the northwestern JES in April,
in agreement with previous investigations (Fig. 11a). Fall blooms initiates in the Tsushima Basin and along the JES
coasts from September to early October and in the central basin of the JES from late October to early November
(Fig. 11b). Spring blooms are terminated generally in the southwestern JES on 11 April ± 9 days and subsequently
in the northeastern JES on 29 May ± 9 days (Fig. 11c). Fall blooms generally terminate on 29 November ± 12 as a
result of the rapidly decreasing PAR beginning in late November (Fig. 11d).
Fig. 12 shows the initiation and termination timing of spring blooms along the JES's northwestern coast during
1998–2019. Spring blooms were initiated earlier in 2002, 2010, 2011 and 2014 and later in 1999, 2005, 2006, and
2013; and were terminated earlier in 2002, 2003, 2010 and 2011 and later in 1999, 2000, 2003, and 2006, identified
by a threshold value of one standard deviation beyond the climatological mean initiation/termination timing (Fig.
12a). The interannual variability in initiation timing is jointly controlled by PAR, stratification, and sea ice melting,
i.e., higher PAR, stronger stratification, and earlier sea ice melting are favorable for earlier spring blooms (Fig. 12b-
d)). The termination and initiation timing of fall blooms are significantly correlated, with a correlation coefficient of
0.74 ($p<0.01$), suggesting that the consumption of upper-layer nutrients determines spring bloom termination, as
supplementary nutrients are rarely available from deeper layers. Additionally, the initiation/termination timings
show trends of occurring earlier by 0.37/0.45 days per year during 1998–2019, coinciding with the increasing trends
in PAR and stratification identified in March and April.
Fall blooms were initiated earlier in 1999, 2015, and 2019, in accordance with strong winds and weak
stratification; and were initiated later in 2003, 2006, 2008 and 2010, when weak winds and strong stratification
occurred. The fall blooms were terminated early in 2011, 2012, 2017 and 2018 and late in 1998, 1999, 2000, and
2015 (Fig. 13a). In comparison to the significant positive correlation observed between the termination and initiation
timing of spring blooms, the corresponding correlation coefficient of fall blooms is only 0.21 ($p>0.1$). In addition,
the initiation/termination timing show trends of occurring earlier by 0.49/1.28 days per year during 1998–2019,
which may be related to the intensified/weakened wind speeds during the fall bloom development/decay periods,
respectively (Fig. 13b).
As shown in Fig. 14a, for spring bloom along the JES's northwestern coast, the termination timing anomalies are
positively correlated with the initiation timing anomalies ($R = 0.75$, $p < 0.01$), while duration anomalies are
negatively correlated with the initiation timing anomalies ($R = -0.72$, $p < 0.01$). The correlations show that the SSC
anomalies from March to May are negatively correlate with initiation ($R = -0.50$, $p < 0.05$) and termination ($R = -$
0.55, $p < 0.01$) timing anomalies, but not statistically correlated with duration anomalies ($R = 0.16$, $p > 0.1$). In
comparison, the impact of AO is more important than that of ENSO for spring bloom timing on the interannual time
scale, because the spring blooms along the JES's northwestern coast would occur earlier ($R = -0.47$, $p < 0.05$) and be
more prolonged ($R = 0.37$, $p < 0.1$) during positive AO, and would terminate earlier during El Niño events ($R = -$



0.42, $p < 0.1$). For fall bloom in the deep Japan Basin, duration anomalies are mainly affected by the initiation
timing anomalies with a negative correlation coefficient of -0.57 above 99% confidence level (Fig. 14b). In addition,,
on interannual time scale, the bloom magnitude is negatively correlated with the initiation timing with a value of -
0.73 ($p<0.01$), and positively correlated with bloom duration with a value of 0.78 ($p<0.01$). Furthermore, the climate
modes also have little effect on modulating the interannual variability of fall bloom timing.
**4 Discussion**
The JES is known as a mid-latitude "Miniature Ocean" in which multiscale oceanic dynamical processes (e.g.,
cross-basin warm and cold currents, upwelling, oceanic fronts, mesoscale eddies, and near-inertial oscillations) and
sea ice occur. The marine ecosystems of the JES are influenced by these complicated processes. The phytoplankton
concentrations can be evidenced by the SSC variability. Previous investigations have revealed bimodal
phytoplankton concentrations blooms occurring in spring and fall. These blooms can be explained by the critical
depth hypothesis, which emphasizes the roles of sea surface wind speeds and temperatures (Chiswell et al. 2013;
Kim et al., 2000; Kim et al. 2007; Taylor and Ferrari, 2011; Yamada et al., 2004). In this study, we revisit the spring
and fall phytoplankton blooms and their mechanisms based on a newly released high-resolution satellite-derived
SSC product.
On interannual time scale, the bloom magnitude anomaly is not statistically correlated with bloom duration
anomaly in spring. This lack of correlation can be attributed to different control factors affecting the durations and
magnitudes of spring blooms. There is adequate light in the entire JES during the spring bloom period, while strong
stratification prevents the upward transport of deep, nutrient-rich waters to supply the upper layer. Therefore, the
bloom duration is mainly controlled by the consumption rate of the accumulated nutrients in the upper layer. SSCC
is a phytoplankton biomass indicator that is related to the amount of accumulated nutrients. Both the phytoplankton
biomass and the bloom duration are positively correlated with the accumulated nutrients. However, more
phytoplankton biomass leads to faster nutrient consumption. Thus, this relation tends to result in a negative
correlation between the durations of and SSCC anomalies associated with spring blooms. As a result, no significant
correlation is found between the spring bloom durations and SSCC anomalies
However, for fall bloom, the interannual variability of bloom magnitude is significantly correlated with the
interannual variability of bloom duration. In comparison with spring bloom, nutrients are limited during the fall
bloom period. The growth of phytoplankton biomass depends on the nutrients supplied by the vertical transport of
deep waters through dynamic oceanic processes. Moreover, these nutrients are consumed immediately after being
transported to the upper layer, and thus, the consumption rate of the nutrients can be ignored. Persistent and active
oceanic dynamic processes tend to transport more nutrient-rich waters to the upper layer, favoring both increased
SSCCs and prolonged fall bloom durations.
Relative to the AO, The interannual bloom magnitude and timing anomalies in the JES are not statistically
correlated with the ENSO, although some existing studies have suggested that El Niño events favor earlier spring
blooms by comparing SSCs recorded in typical El Niño years (e.g., 1998 and 2002) with those representing normal
or La Niña years (e.g., 1999 and 2001) (Yamada et al., 2004; Yoo and Kim, 2004). Indeed, the JES is influenced by





ENSO at interannual time scales (Cheon, 2020; He et al., 2017; Son et al. 2016; Wang and Chan 2002; Wang et al.
2000). However, ENSO event-induced anomalies somehow result in contradictory SSC responses in the JES. For
instance, El Niño events tend to be followed by weak wind speeds that favor early spring blooms (Yamada et al.,
2004). On the other hand, El Niño events tend to co-occur with warm winters and thus less sea ice, thus favoring
negative spring SSC anomalies in the JES (Hong et al., 2001; Park et al., 2014). In terms of fall blooms, higher
SSCs and longer bloom durations are expected, as active typhoons benefit the vertical transport of nutrient-rich
waters during El Niño years (Goh and Chan, 2010). However, at the same time, the warmer background states that
occur during the summer-fall seasons in the JES act to inhibit SSC increases and thus cancel out the role of typhoons
(Cheon, 2020).
However, some issues remain unclear, including but not limited to (1) obtaining a quantitative assessment of the
driving factors that account for JES SSC variabilities; (2) the relation between JES SSCs and climate change; and (3)
the JES SSC variabilities induced by changes in the phytoplankton community structure and the interactions
between grazers and phytoplankton. Especially, the complex predator-prey interactions are crucial to the temporal
changes in phytoplankton concentrations (Behrenfeld, 2010; Behrenfeld and Boss, 2017). Thus, these pending issues
must be further investigated through interdisciplinary collaboration.
**5 Summary**
In this study, we investigate the spring and fall SSC blooms and their interannual variability by employing high-
resolution satellite remote sensing products. The new findings are summarized as follows.
(1) On the annual cycle, the influences of local oceanic dynamic processes (e.g., upwelling, oceanic fronts,
mesoscale eddies, and near-inertial oscillations) on the bloom magnitude and timing of the entire JES are limited,
compared with the PAT and stratification.
(2) The interannual variability in the JES SSC consists of spring and fall components, which occur along the
JES's northwestern coast and in the deep JES basin, respectively. Stronger PAR and stratification favor positive SSC
anomalies of spring bloom, whereas weaker stratification favors positive SSC anomalies of fall bloom.
(3) For spring bloom, the interannual variations in initiation timing have crucial effect on the interannual
variations in termination timing, duration and magnitude. However, the interannual variability of bloom magnitude
is not statistically correlated with that of bloom duration. Additionaly, during positive AO, the bloom magnitude
would increase, and spring blooms along the JES's northwestern coast would occur early and be prolonged. During
El Niño events, spring blooms tend to terminate early.The spring bloom initiation/termination timing has shifted
earlier by 0.37/0.45 days annually along the JES's northwestern coast.
(4) For fall bloom, duration anomalies are mainly affected by the initiation timing anomalies with a negative
correlation coefficient of -0.57. Moreover , the bloom magnitude is significantly correlated with the initiation timing
bloom duration on interannual time scale. However, the climate modes also have little effect on modulating the
interannual variability of fall bloom timing, since fall blooms are dominated by oceanic dynamic processes. The fall
blooms initiation/termination timing has shifted 0.49/1.28 days earlier annually in the deep Japan Basin.



(5) Relative to the AO, the interannual variability in JES SSCs are not statistically correlated with ENSO, which
tends to induce atmospheric and/or oceanic anomalies that favor a counterbalanced response in SSC.
*Code and data availability.* The SSC data are available at https://resources.marine.copernicus.eu. The *in situ*
chlorophyll-a concentration data are available at https://www.ncei.noaa.gov/products/world-ocean-database. The
PAR and *k* data are available at http://www.globcolour.info/. The SIC data are available at http://nsidc.org/. The
SSH and sea surface geostrophic current anomalies data are available at http://www.aviso.altimetry.fr/duacs. The
SST data are https://resources.marine.copernicus.eu. The surface wind vectore data are available at
https://cds.climate.copernicus.eu/cdsapp#!/dataset/reanalysis-era5-single-levels?tab=form. The monthly Niño3.4 and
AO indices are available at http://climexp.knmi.nl/selectindex.cgi?id=someone@somewhere. The WOA18 data are
available at https://www.ncei.noaa.gov/access/world-ocean-atlas-2018.

*Author contributions.* **Dingqi Wang**: Writing – original draft, Writing – review & editing, Methodology, Software.
**Guohong Fang**:Writing – review & editing, Supervision. **Shumin Jiang**:Writing – original draft, Methodology.
**Qinzeng Xu**:Writing – review & editing. **Guanlin Wang**:Writing – review & editing. **Zexun Wei**:Writing – review
& editing, Funding acquisition. **Yonggang Wang**: Writing – review & editing. **Tengfei Xu**: Writing – review &
editing, Resources, Funding acquisition.

*Competing interests.* The contact author has declared that neither they nor their co-author has any competing
interests.

*Disclaimer.* Publisher's note: Copernicus Publications remains neutral with regard to jurisdictional claims in
published maps and institutional affiliations.

*Financial support.* This study is jointly supported by the National Key Research and Development Program of
China (Grant No. 2020YFA0608800), and the National Natural Science Foundation of China (Grant No. 41821004).
This work got the data service support from the Marine Environment Data Service System which supported by the
National Key Research and Development Program of China (2019YFC1408405).

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

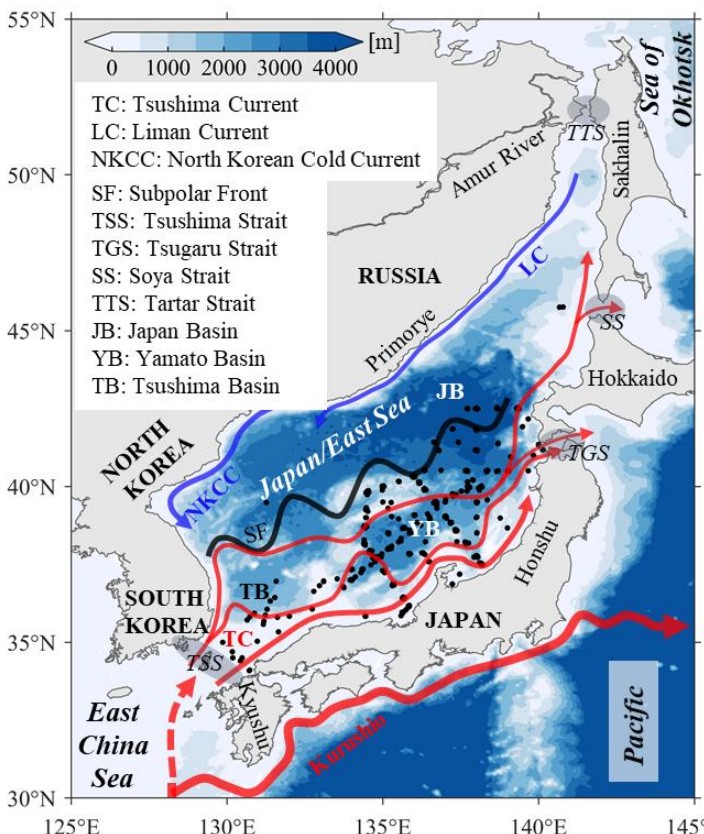

**Figure 1. Topography of the JES. Red/blue arrows represent warm/cold currents (Yabe et al., 2021). Black dots indicate**
**the *in-situ* observation stations of chlorophyll-a concentration, as derived from the WOD18.**






**Figure 2. Regression between the *in-situ* observed chlorophyll-a concentrations from WOD18 (x-axis) and satellite-derived (y-axis) sea surface chlorophyll-a concentrations (SSCs).**

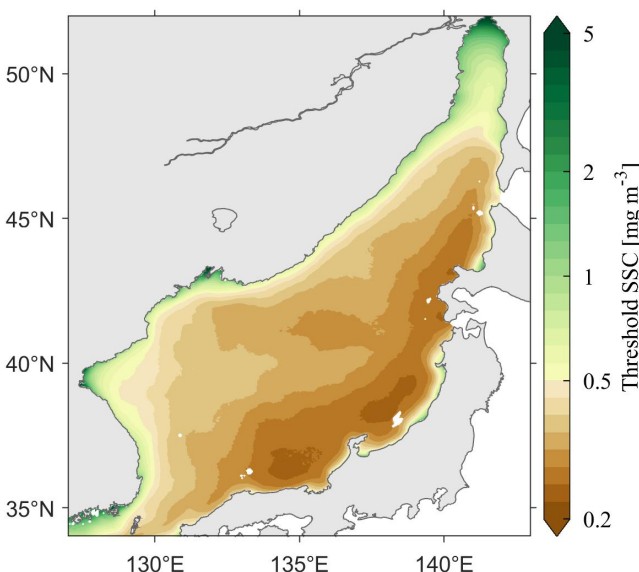


**Figure 3. Distribution of threshold SSC for a bloom at each grid in the JES over 1998–2019.**






**Figure 4. Distribution of climatological monthly mean SSC in the JES over 1998–2019. The black contours indicate the threshold criterion used to identify SSC blooms for the entire JES.**

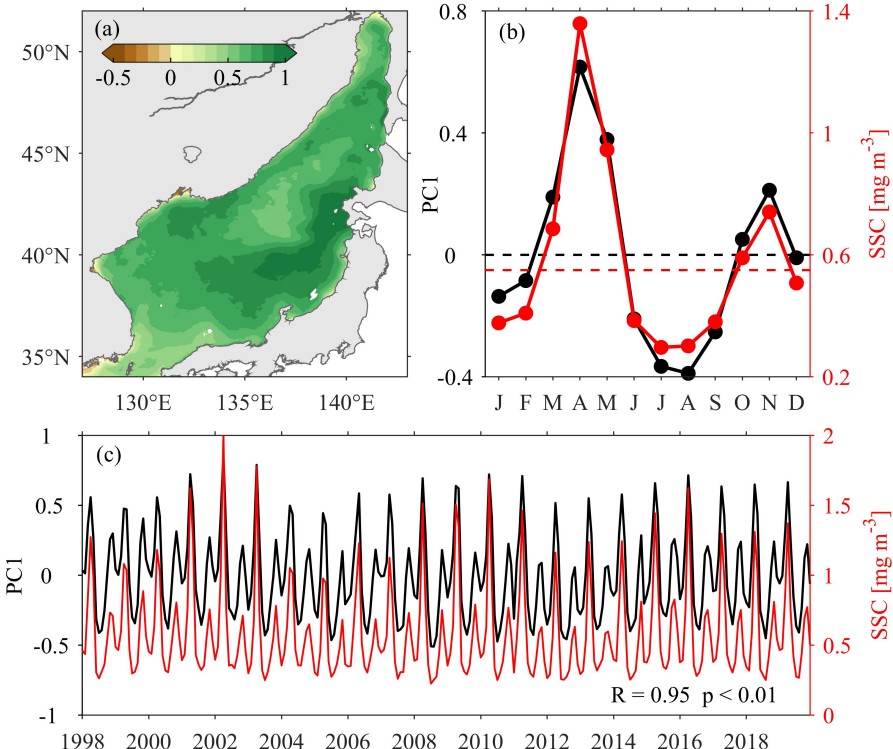

**Figure 5. EOF analysis of the monthly mean SSC in the JES: (a) spatial pattern of the first EOF mode, (b) climatological and (c) monthly time coefficients (black lines). The red lines in (b) and (c) represent climatological and monthly mean area-averaged SSC values in the JES, respectively. In (b), the black dash line indicates the zero , while the red dashed line indicates the threshold criteria (0.55 mg m⁻³) of SSC blooms for the entire JES, respectively.**

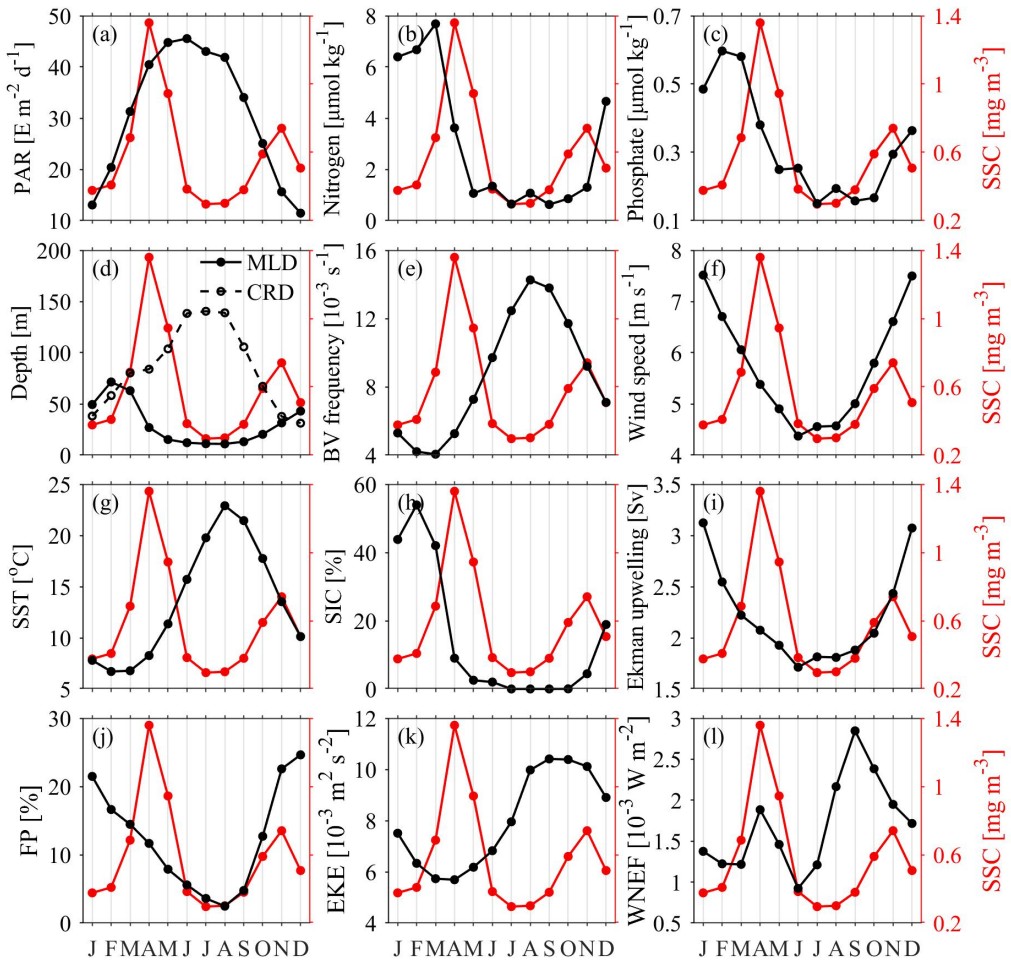

578

**Figure 6. Climatological monthly mean series of area-averaged physical environmental factors (black lines) and area-averaged SSCs (red line): (a) photosynthetically active radiation (PAR); (b) nitrogen; (c) phosphate; (d) mixed layer depth (MLD; solid line) and critical depth (dashed line); (e) Brunt Väisälä (BV) frequency; (f) wind speed; (g) sea surface temperature (SST); (h) sea ice concentration (SIC); (i) frontal probability (FP); (j) Ekman upwelling transport; (k) wind-induced near-inertial energy flux (WNEF); and (l) eddy kinetic energy (EKE).**



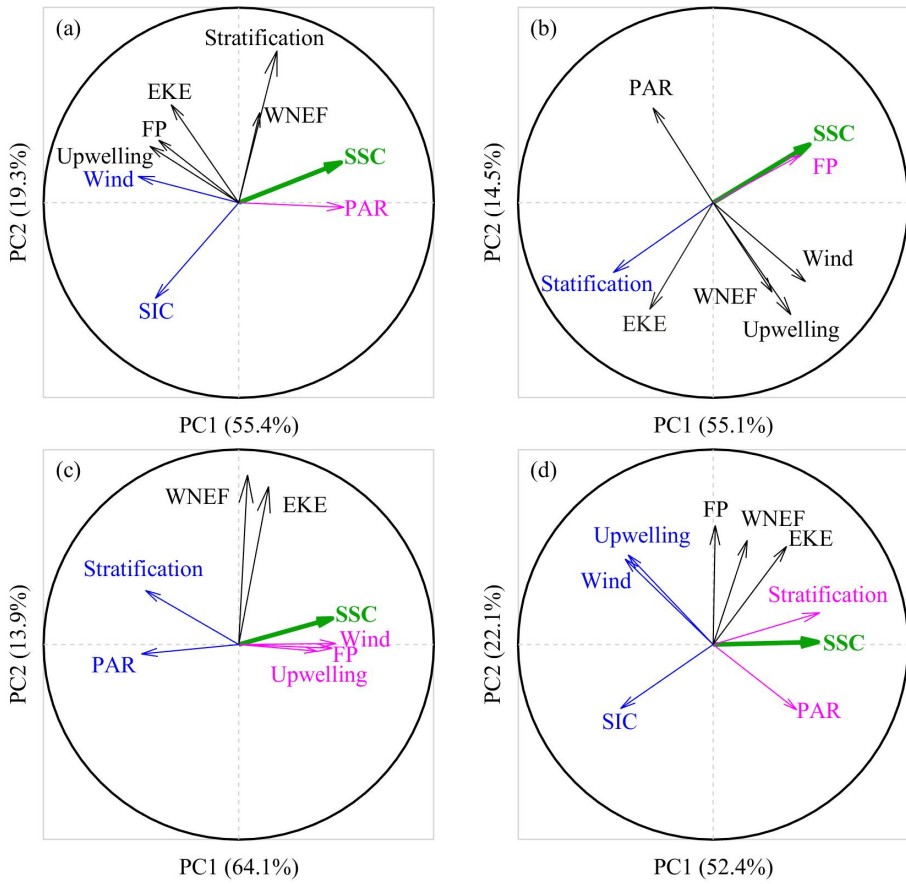

584
**Figure 7. PCA analysis of physical environmental factors (arrows) during different SSC evolution stages: (a) raise (January–April) and (b) decline (April–July) stages of spring blooms; (c) raise (July–November) and (d) decline (November–January) stages of fall blooms. The x- and y-axes represent the first and second principal components (PC1 and PC2), with their variance contributions marked as percentages in brackets of the x- and y-labels respectively. Arrows close to each other denote that the corresponding factors are positively correlated, whereas those are symmetrically opposed to each other suggest they are negatively correlated. Orthogonal arrows mean they are not correlated with each other. The projections of the arrows on the x- and y-axes indicate the extent to which the factors are explained by PC1 and PC2, respectively. The factors positively (negatively) correlated with SSC (green thick arrows) are highlighted with purple (blue) arrows.**

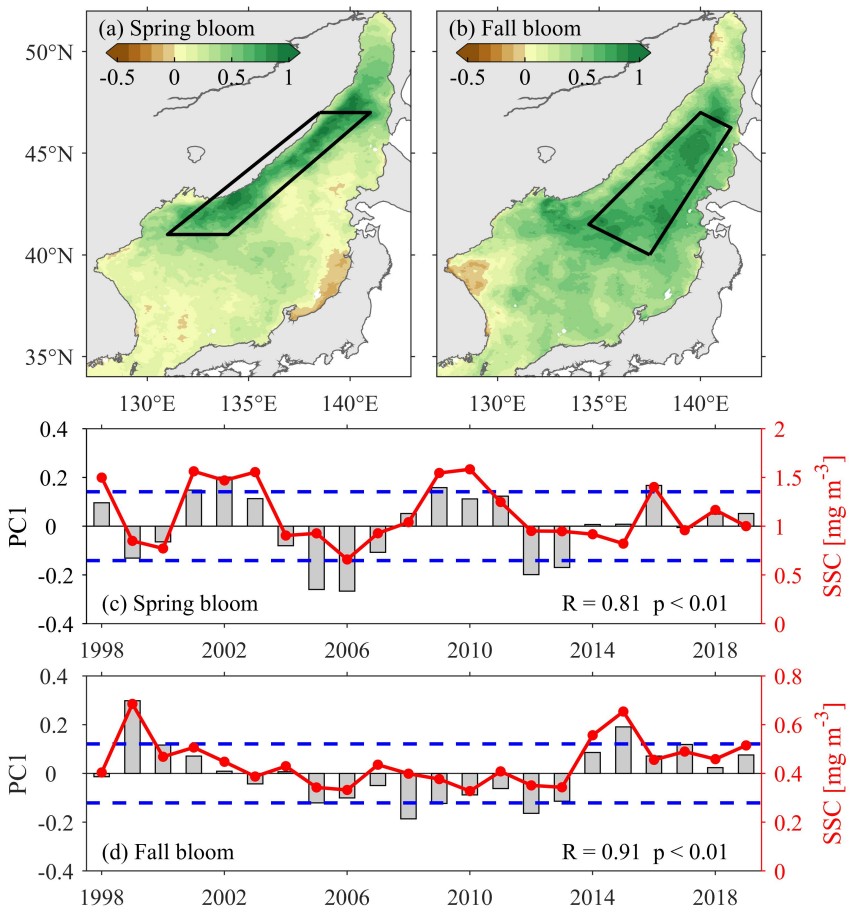

594

**Figure 8. EOF analysis of SSC anomalies in the JES. Spatial patterns of the first EOF mode (EOF1) during (a) spring**
**(March–May) and (b) fall (October–November). The corresponding time coefficients (grey bars) of EOF1 with standard**
**deviation (blue dash lines), and area-averaged SSC (red lines) during (c) spring and (d) fall. The areas averaged are**
**marked with black boxes in (a) and (b), respectively.**



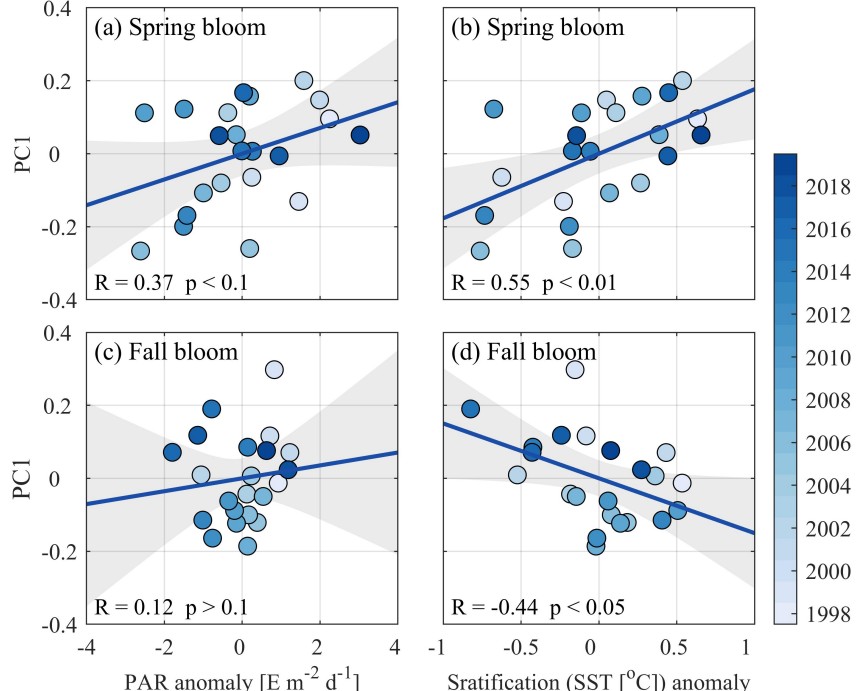

**Figure 9. Scatterplots (colored dots) and linear fitting (blue lines) between PAR anomalies and PC1 in (a) spring and (c) fall and between stratification anomalies and PC1 in (b) spring and (d) fall. The gray shading indicates the 95% two-sided confidence bounds.**





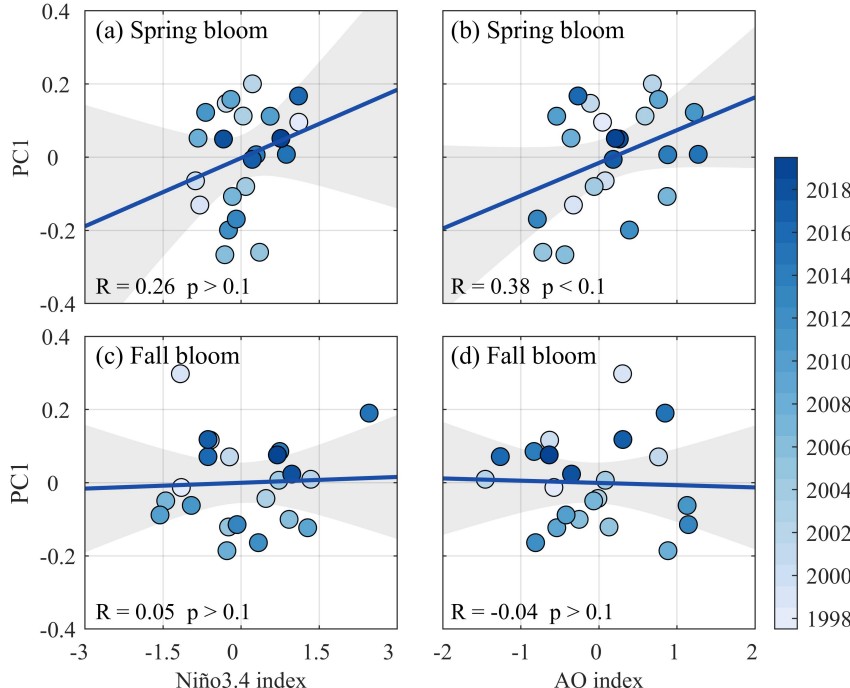

603

**Figure 10. Scatterplots (colored dots) and linear fitting (blue lines) between Niño 3.4 index and PC1 in (a) spring and (c)
fall and between AO index and PC1 in (b) spring and (d) fall. The gray shading indicates the 95% two-sided confidence
bounds.**



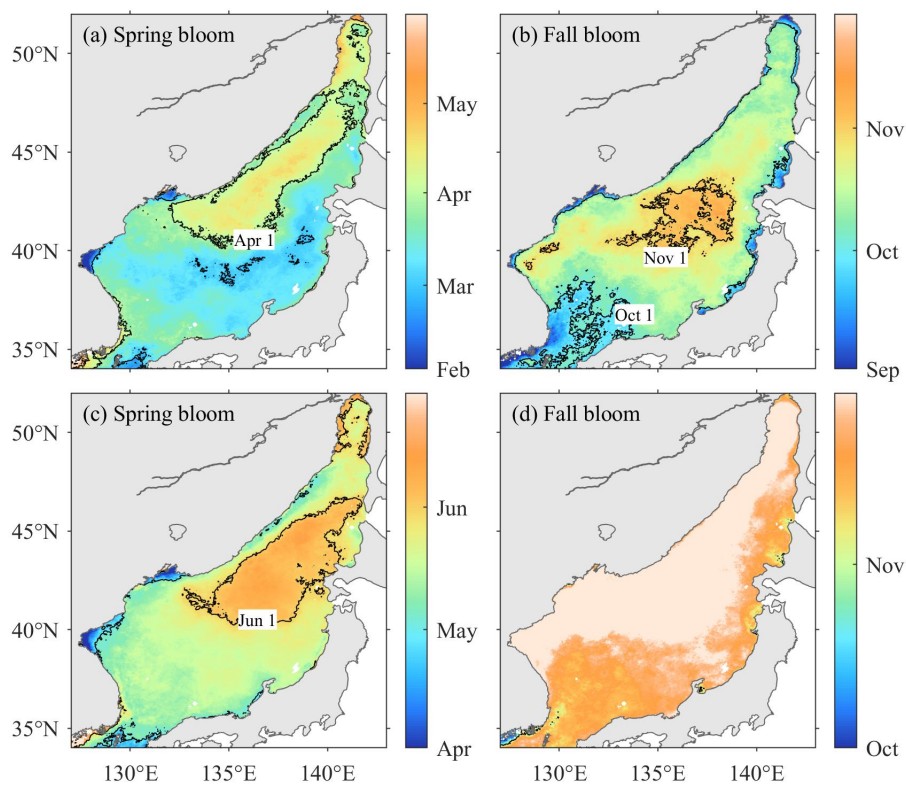

607

**Figure 11. The climatological distributions of the initiation timing of (a) spring and (b) fall blooms; and the termination timing of (c) spring and (d) fall blooms. The interval between black contours is one month.**



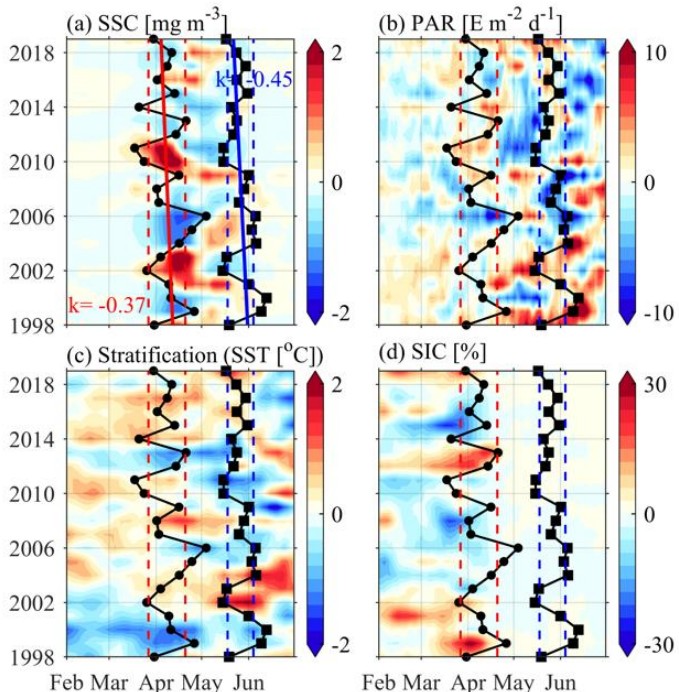

**Figure 12. (a) SSC, (b) PAR, (c) stratification, and (d) SIC anomalies averaged along the JES's northwestern coast (marked by the black box in Fig. 8a) from 1998 to 2019. The spring bloom initiation and termination timing are represented by solid dots and square lines, with the solid red/blue lines indicating the linear trends, and the dashed red/blue lines indicating the one standard deviation beyond the climatological mean initiation/termination timing, respectively.**

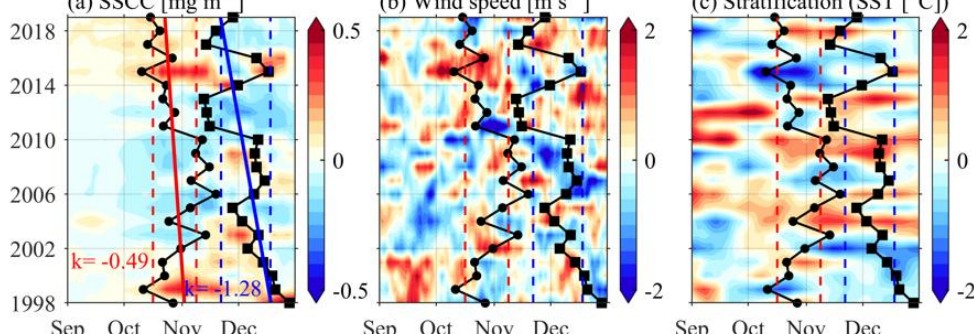

**Figure 13. (a) SSC, (b) PAR, and (c) stratification anomalies averaged in the deep Japan Basin (marked by the black box in Fig. 8b) from 1998 to 2019. The fall bloom initiation/termination timing are represented by solid dots/square lines, with the solid red/blue lines indicating the linear trends, and the dashed red/blue lines indicating the one standard deviation beyond the climatological mean initiation/termination timing, respectively.**



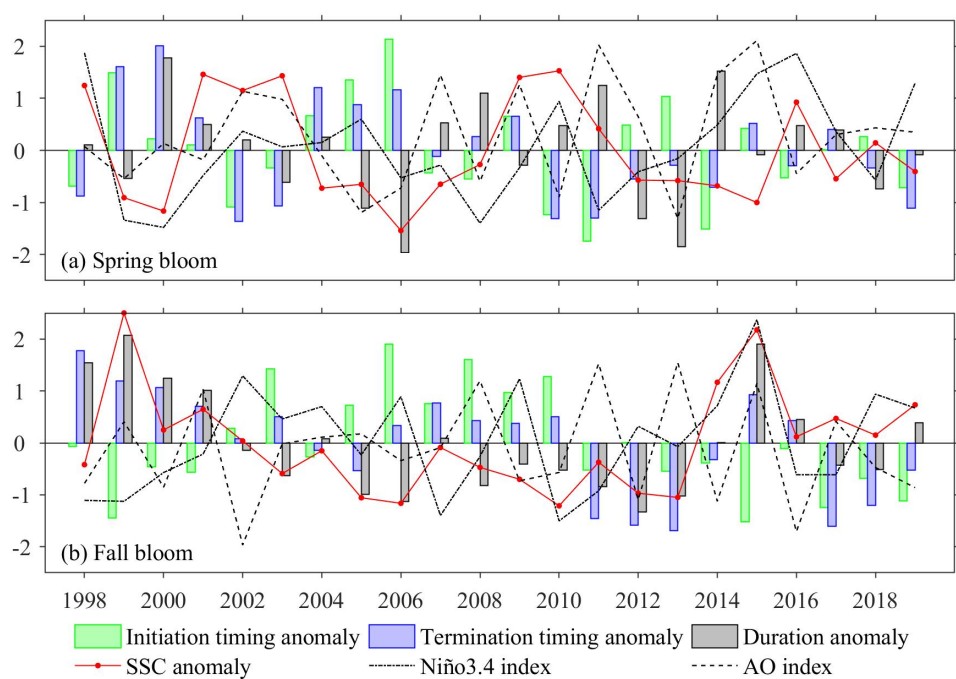

**Figure 14. Time series of initiation timing anomalies (green bars), termination timing anomalies (blue bars), duration anomalies (gray bars), area averaged SSC anomalies (red lines), Niño 3.4 index and AO index for (a) spring, and (b) fall blooms. The averaged areas are marked by black box in Fig. 8a and 8b for spring and fall blooms, respectively. The time series of all variables are normalized, i.e., divided by their corresponding standard deviations (STD).**