# Peer review of "Satellite-detected sea surface chlorophyll-a blooms in the Japan/East Sea: magnitude and timing"

_EGUsphere, 2022_

## Author Comment (AC1)

The study of Satellite-detected sea surface chlorophyll-a blooms in the Japan/East Sea: magnitude and timing by Wang et al. applied satellite observations over 20 years for identifying the chlorophyll bloom in the Japan/East Sea. By comparing with all the major physical parameters, e.g., wind, eddies and fronts, they find the impact of solar radiation and stratification are actually more important to determine the bloom of phytoplankton. The presented information is interesting, but the scientific soundness should be further confirmed. In particular, the satellite observations are limited in the surface, but the nutrient supply at subsurface is also predominant. A major revision is necessary for presenting the credibility of their conclusion and improving the description.

**Major comments:**

The dynamical dependence between interannual index and regional chlorophyll should be further investigated. It is not surprising to find some statistically significant correlation, but the underlying mechanisms should be further explored. The authors tried to present a dependence between ENSO and chlorophyll bloom via the intensity of Tsushima Warm Current. If this is the case, the intensity of the current should be added for presenting a comprehensive relationship. In particular, the lag among ENSO, warm current, front, chlorophyll is of great interesting.

Thanks for the reviewer's comments. Based on the analysis of several spring bloom cases, Yoo and Kim (2004) suggests that ENSO events could modify the location and maintenance of subpolar fronts by influencing the intensity of Tsushima Warm Current, and eventually change initiation region and timing of the spring bloom. Based on this result, we treated ENSO events as a candidate to affect the chlorophyll bloom and try to check its effects statistically. However, the correlation results suggests that there is not a clear dependence between ENSO and chlorophyll bloom, as shown in Fig.10 in the original version of paper. Thus, we suggest that the ENSO events' effects are not as simple as suggested in case studies, and could be covered by the interaction of different physical processes. Therefore, the correlation between the ENSO and warm current, fronts, chlorophyll is not included in the paper, and the figures can be find below.

As suggested by the reviewer, here we calculated the lag correlation between ENSO and Tsushima Warm Current transport anomalies (TUSA), between TUSA and front probability anomalies (FPA), between TUSA and SST anomalies (SSTA), between TUSA and sea surface chlorophyll-a concentration anomalies (SSCA), which are the possible action steps of TUSA on the SSCA, as suggested by Yoo and Kim (2004). As shown in Fig. 1, each one of the results is not significantly correlated at any given lagging time, suggesting the SSCA is more likely to be affected by local physical processes, rather than ENSO events.

**Figure S1.** The lag correlations between (a) Tsushima warm current transport anomalies (TUSA) and Niño3.4 index; (b) front probability anomalies (FPA) and TUSA; (c) area averaged sea surface temperature anomalies (SSTA) and TUSA; (d) area averaged sea surface chlorophyll-a concentration anomalies (SSCA) and TUSA. The red dashes represent the 95% confidence level.

Highly similar method has been formerly applied in other oceans, e.g., the South China Sea. But they presented more robust features with intercorrelation at seasonal/semiseasonal and interannual variability that the authors should consider to implement in this study. In particular, the seasonal/semiseasonal cycles are usually prominent for all the parameters (Legaard and Thomas, 2006) and a significant correlation can be achieved all the time by adding a lag. It is more meaningful to explore the dependence at interannual variability after removing the seasonal/semiseasonal cycle. Thanks for the reviewer's comments. We have reduced the seasonal/semiseasonal cycles in the analysis of interannual variability. Figure 2 shows the first EOF mode (EOF1) of SSC anomalies in the JES, which contributes 22.4% of the total variance. The occurrence frequencies of the peaks of PC1 over months are shown in Figure 2c. The SSC anomalies show more peaks in April and October, coinciding with spring and fall blooms. Therefore, in the manuscript, we show the EOF analyses for the interannual SSC anomalies (remove seasonal variability) during spring (March–May) and fall (October–November), respectively, in order to remove the seasonal and semi-annual signals. As shown in Figure 3, lag correlations show that both the ENSO and AO have insignificant correlation with the interannual variability of fall bloom magnitude.

**Figure 2.** Empirical orthogonal function (EOF) analysis results of monthly mean SSC anomalies in the Japan/East Sea (JES). (a) Spatial pattern of the first empirical orthogonal function (EOF) mode (EOF1); (b) the 13-month running mean time coefficients of EOF1 (PC1); (c) the peak numbers of PC1 in different months.

---

## Author Comment (AC2)

We would like to thank the Editor and Reviewer for their nice help on our manuscript. We have revised the manuscript accordingly, and reply of the comments are given one by one as follows.

*Wang et al. analyzed long term variation of satellite chlorophyll in JES. Although the authors put together many kind of different data, the descriptions were fairly simplify for this complex ecosystem with large spatial variation. The discussion (many are in the results) is very poor and speculative, ignoring previous studies. Unfortunately, I cannot recommend the publication.*

Reply: We appreciate the Reviewer for his/her detailed comments on our manuscript. However, these comments, or, most of these comments are totally not acceptable to us. Arguments are replied one by one as follows.

1. *The analysis of the interannual variation was only limited to the northwest coastal area for spring and deep Japan Basin for fall. Although it is based on the high variation of 1$^{st}$ mode of EOF, those areas are only limited areas of the JES, and I do not think it is suitable to conclude it is applicable to the whole JES.*

Reply: EOF is a commonly used method that can extract temporal-spatial variation signals over a region, i.e. for a 3-D matrix $V(i,j,k)$, where $V$ is the variable, and i, j indicate its zonal, meridional direction, and k indicates the timeline. For its spatial pattern $EOF(i,j)$, values with larger absolute values indicate that there are significant variations within these area; On the contrary, the area with small absolute values means it does not exhibit significant variations. Taken ENSO as an example, it is the most dominant mode in the Pacific Ocean, but its variations are limited only within the equatorial central-eastern Pacific Ocean. The analysis of JES Chl-a is also like this, say, the dominant interannual variability of JES SSC is limited along the JES's northwestern coast, and in the deep Japan Basin, for spring and fall, respectively (Fig 8a-b), and the interannual variability in the rest of the JES are much smaller and not significant compared with the above two identified areas. Therefore, it can be solidly concluded that in terms of the interannual variations of JES SSC, it is appropriate to focus on the areas that along the JES's northwestern coast, and in the deep Japan Basin, for spring and fall, respectively.

2. *As the results correlations are OK, but the just correlation cannot explain the cause and effect. I think authors deeply concern about the cause and effect seriously.*

Reply: We totally agree with the Reviewer that deeply investigated are needed to further explain the mechanisms. However, we also have to admit that it is not achievable to reveal the detailed cause and effect, unless we obtain time series of vertical profiles of nutrient, Chl-a, temperature, salinity,

and velocity in the JES, and to build an ocean dynamics-ecosystem coupled numerical model to conduct sensitivity experiments, both of which are far beyond scope of this manuscript. Instead, in this study, we we focus on revealing the favorable/restricting factors during the SSC raise and decline stages, and investigating the variations in bloom magnitude and timing in spring and fall, as well as their related physical environmental factors and climate modes, based on recently released high-resolution data of SSC, SST, SSW (sea surface wind), etc. The possible linkage between these marine dynamical factors and SSC, in fact, have been discussed by previously studies (Yamada et al., 2004; Yamada and Ishizaka, 2006; Kim et al., 2007; Jo et al., 2014; Lee et al., 2015; Ishizaka and Yamada, 2019), and have reviewed and discussed in the introduction and discussion sections in the manuscript. One of the gaps filled by this study could be given by comparing with Yamada et al. (2004).    Published in the Progress in Oceanography, Yamada et al. (2004) have done well job by studying the seasonal and interannual variability of sea surface chlorophyll a concentration in the Japan/East Sea (JES). However, since there was limited and coarse data of SSC (1997–2002), monthly mean data derived by weekly and/or 8th-day data, 9 km horizontal resolution) used in Yamada et al. (2004), the bloom timing (initiation/termination timings, and durations) cannot be well identified. Their discussion of the interannual variability in JES SSC only gives comparison between the 1997/98 El Niño and 1999/00 La Niña events, without showing statistical relationship between the SSC and ENSO. Therefore, in this study, we further investigated the seasonal and interannual variations in JES SSC, with focus on the magnitudes and durations of the JES SSC blooms, based on the newly released high resolution SSC products (1998–2019) at a 4-km resolution with daily time interval). Some new findings are:

(1) Increased photosynthetically active radiation (PAR), weakened wind speeds and melting sea ice induce the spring bloom raise stage; enhanced stratification causes the decline stage. Destratification and active dynamic processes induce fall bloom development, while decreased PAR and increased sea ice cause fall bloom decay.

(2) The interannual variability in the JES SSC consists of spring and fall components, which occur along the Russian-Korean coasts and in the deep JES basin, respectively. Stronger PAR and stratification favor positive SSC anomalies of spring bloom, whereas weaker stratification favors positive SSC anomalies of fall bloom.

(3) The interannual variability in JES SSCs are not statistically correlated with ENSO, because of these climate events tend to induce atmospheric and/or oceanic anomalies that favor a counterbalanced response in SSC.

(4) No significant correlation exists in spring between interannual bloom duration and bloom magnitude variabilities, but a significant positive correlation exists in autumn when bloom timing and SSC are both dominated by active oceanic dynamical processes.

*3. Discussion is very weak, and almost nothing was discussed about the reviews in the introduction.*

Reply: We have tried our best to discuss the favorable and restrict factors for the spring and fall blooms, and explain why the interannual variations of JES SSC are not statistically correlated with interannual climate modes such as ENSO, albeit ENSO should have impacts on the JES SSC by modulating oceanic and atmospheric factors in the JES, which in turn influence JES SSC. Indeed, the discussion should be the deeper the better, and we would greatly appreciate to the reviewer and make revision accordingly, if he/she could give suggestions and comments clearly.

*4. It may be a good idea to show correlations with Table(s).*

Reply: We are not sure about which correlations are better to show in tables. We prefer to show correlations by figures is because it can include more information with less pages than tables.

*Abstract*
*19 Delete "However".*
Reply: Deleted in the revised manuscript.

*19-21 I do not think the authors really showed PAR and stratification is more important than other processes.*

Reply: Our analyses show that the PAR and stratification are more important than other processes at seasonal and interannual time scales, as shown in Figure 7, Figure 9, Figure 12 and Figure 13. This conclusion is also in agreement with hypothesis from previous works. Since we have not analyzed JES SSC variation at other time scales such as intraseasonal and decadal time scale, we revised this sentence as:

> The influences of local oceanic dynamic processes (e.g., upwelling, oceanic fronts, mesoscale eddies, and near-inertial oscillations) on the bloom magnitude and timing of the entire JES are not critical, compared with the photosynthetically active radiation (PAR) and stratification at seasonal and interannual time scales.

*21 PAT should be PAR, and it need to be defined.*

Reply: Revised.

*21 'In addition" may not be a good connection here.*

Reply: Deleted.

*23-24 Timing affect magnitude?*

Reply: We are sorry for the misleading expression. The abstract has been re-written:

No significant correlation exists in spring between interannual bloom duration and bloom magnitude variabilities, but a significant positive correlation exists in autumn when bloom timing and SSC are both dominated by active oceanic dynamical processes.

*24-25 In the text, it was said ENSO is not important?*

Reply: ENSO may be important for JES SSC variability. However, the interannual JES SSC variabilities are not statistically correlated with El Niño-Southern Oscillation (ENSO), because of ENSO event-induced anomalies tend to result in contradictory SSC responses in the JES.

*25 Duration is mainly affected by initial timing?*

Reply: We are sorry for the misleading expression. The abstract has been re-written:

No significant correlation exists in spring between interannual bloom duration and bloom magnitude variabilities, but a significant positive correlation exists in autumn when bloom timing and SSC are both dominated by active oceanic dynamical processes.

*26 Duration and initial timing have significant influence on the bloom magnitude?*

Reply: We are sorry for the misleading expression. The abstract has been re-written:

No significant correlation exists in spring between interannual bloom duration and bloom magnitude variabilities, but a significant positive correlation exists in autumn when bloom timing and SSC are both dominated by active oceanic dynamical processes.

*1. Introduction*
*78 SSC "is" related*

Reply: Revised.

*79 "based on composite analyses without test of confidence level"   It is not very clear what is the problem.*

Reply: We are sorry for the misleading expression. This sentence has been deleted and the introduction has been re-written.

*89 I am not sure previous analysis really did not focus on the whole area and this study did.*

Reply: We checked the relevant English literature and found that there is indeed a lack of a comprehensive study on bloom magnitude and timing in multiple time scales in the entire JES area. However, there may have been relevant reports in Japanese or Korean journals, which we did not reach. Therefore, we deleted "in the whole JES area" this expression and revised the relevant contents of introduction.

*90 favorable/restricting factors?  You only showed the correlations, and the cause/effect can be discussed.*

Reply: We reveal the favorable/restricting factors during the SSC raise and decline stages of spring and fall blooms by showing the time series, correlations, and PCA analysis. The results and discussion are in section 3 and section 4, not in introduction.

*2. Data and Methods*

*100 Is WOD18 station data?  Is the comparison to satellite data based on daily match-up?*

Reply: Black dots in Figure 1 indicate the *in-situ* observation stations of chlorophyll-a concentration, as derived from the WOD18. The comparison is based on daily match-up.

*103 Did you check if there is no interannual vias?*

Reply: I assume the reviewer meaned "bias" when mentioned "interannual vias". The satellite SSC data is derived from ocean color, so it may be not reliable and thus needs validation by comparing with in-situ observation of Chl-a concentration, before used for investigation. Since the in-situ observation of Chl-a concentration are in shortage, the validation of satellite data is done by regressing the satellite SSC onto the 10 m Chl-a in WOD18. The high correlation between the two data indicates that the satellite SSC are generally in good agreement with those of in-situ observed Chl-a in WOD18, and can be used for further investigation of JES SSC variability, as shown in Fig. 2.

*105 Are PAR and k data daily, and later make monthly composite?*

Reply: PAR and *k* data are daily products, as shown in Table 1. In the later analysis, PAR data is averaged monthly. The daily *k* data is used to calculate the daily critical depth (CRD), which is then averaged monthly.

*119 Add "climatology" for WOA18 temporal coverage.*
Reply: Revised.

*124-126 "To identify blooms, the threshold is about 0.55"   I do not understand this sentence. Is the value spatial average?*
Reply: The threshold 0.55 mg m$^{-3}$ is "calculated from the **area-averaged** SSC data" (see Line 125 in the initial manuscript). The definition of this threshold value is adopted from Maúre et al. (2017).

*155 What is "an adaptive data analysis technique"?*
Reply: Large datasets are increasingly common and are often difficult to interpret. Principal component analysis (PCA) is a technique for reducing the dimensionality of such datasets, increasing interpretability but at the same time minimizing information loss. It does so by creating new uncorrelated variables that successively maximize variance. Finding such new variables, the principal components, reduces to solving an eigenvalue/eigenvector problem, and the new variables are defined by the dataset at hand, not
*a priori*, hence making PCA an adaptive data analysis technique (Jolliffe and Cadima, 2016). In a simple word, the PCA method could extract signals based on the data itself, without artificial determination. We have revised this sentence as follows:

> The principal components are found by solving an eigenvalue/eigenvector problem wihout *a priori*, hence making PCA an adaptive data analysis technique.

*163 Was logarithmic transform not used before calculating monthly mean of SSC?*
Reply: The daily SSC data is firstly monthly averaged, and then is logarithmically transformed prior to the PCA and EOF analysis.

*3. Results*
*3.1*
*166 "Seasonal variability of bloom magnitude" Bloom is only spring and fall, and this title seems to be strange.*
Reply: We revised this section title as "Seasonal bloom magnitude".

*185 Is the analysis with average of whole JES?*

Reply: Yes, "The seasonal time coefficient of EOF1 coincides with the **area-averaged SSC in the JES**, showing double peaks of 1.36±0.28 and 0.74±0.07 mg m–3 in April and November, respectively" (please see lines 184-185 of the initial manuscript).

*187 Is this shortwave radiation same as PAR?*

Reply: No. Incident PAR on the Earth's surface represents a part of the solar radiation spectrum from 0.4 μm to 0.7 μm that is absorbed, transferred and stored within ecosystems.

*188-220 I think this include too much speculations and sloppy words, and they should move to discussion and discuss carefully. I think the dynamics depends on the regions of the JES.*

Reply: The analyses are based on the seasonal evolutions of different variables that related to the seasonal variability of JES SSC. The conclusions are solid and corroborated by previously investigations, and the underlying mechanisms are common sense that can be easily derived. These analyses are important basis for discussing the dominant and limiting factors affecting the seasonal variability of SSC in the JES (as shown in Figure 7), therefore, it should better insert before Figure 7, rather than in the discussion section.

*202, 205 The sea ice melting effect should be important for very limited area.*

Reply: The sea ice melting effect is important for not very limited area It can influence the area that along the Russian-Korea coasts in spring, which is considerable for JES.

*204 "favors" should be "corresponds"*

Reply: Revised.

*208 "ocean dynamics" is too broad words.*

Reply: We totally agree with the reviewer that "ocean dynamics" is too broad, and well descript the complicated JES, which involves multiple ocean dynamical processes. Here, we used "ocean dynamics" as a sketch, followed by analysis of the specific ocean dynamical processes (e.g., upwelling, oceanic front, eddy, etc.) that may related to the seasonal variability of SSC in the JES.

*208-209 "dominate the upper layer nutrients" It is too general.*

Reply: The factors that influence upper layer nutrients can be classified as two aspects: (1) input fluxes from atmosphere, land runoff, and sea-ice melting; and (2) deeper layer supplements induced by oceanic dynamical processes.

*210 "in accordance with the enhanced upwelling and frontal probability"  It is too general.*

Reply: In Figures 6i and 6j, we can see that after October, nutrients increase in accordance with the enhanced upwelling and frontal probability. We are confused about what means "It is too general".

*211-212 EKE distribution was not shown.*

Reply: Trusenkova (2014) proposed that in the northern JES, where the mean EKE is several times less than in the southern sea, areas of considerable variability are detected (Figure 2 in Trusenkova, 2014). We added this reference to the original manuscript as follows:

> The EKE is larger in the southern JES from August through December, consistent with previous investigation results that the EKE in the southern sea is several times more than that in the southern JES (Trusenkova, 2014).

[Figure]

Reference:

Trusenkova, O. O.: Variability of eddy kinetic energy in the Sea of Japan from Satellite Altimetry data, Oceanology, 54 (1), 8–16, https://doi.org/10.1134/S0001437014010111, 2014.

*212-220 None of those points are shown as the results.*

Reply: We thank for the reviewer's suggest, and these points have been summarized and added in the summary as follows:

JES SSCC is characterized by bimodal blooms in spring and fall. Different physical environmental factors are responsible for the JES SSCC evolution during the raise and decline stages of the spring and fall blooms. Increased PAR, weakened winds and melting sea ice are the controlling factors inducing the initiation of spring blooms. Enhanced stratification, which inhibits the upward transport of nutrient-rich waters to the upper layer to compensate for the rapid consumption of nutrients, is the controlling factor inducing the termination of spring blooms. Destratification and active oceanic dynamic processes are favorable factors for fall blooms because they transport nutrient-rich waters to the upper layer. Declining PAR tends to play a negative role and serve as a limiting factor for phytoplankton growth when it is reduced to a certain extent, thereby resulting in the termination of fall blooms.

*221-240 Is the PCA analysis was conducted with monthly seasonal climatology of the whole spatial average? I think if it spatial average it is very difficult to understand because of the regional difference of the JES.*

Reply: The PCA analysis was conducted with monthly logarithmically transformed SSC data of the whole spatial average. As shown in Figure 5b, the EOF results show consistent seasonal cycle in both the time coefficients (time coefficients represent the most significant seasonal signal of JES SSC) and area-averaged SSC in the entire JES. Therefore, in terms of seasonal variability, we can use the spatial average SSC data in the PCA analysis to reveal the favorable/restrict factors for SSC blooms.

*3.2*
*245 What is the criteria of "significance".*
Reply: We revised "significant" as "large".

*251 "JES SSC" Are these from the black boxed areas?*
Reply: No. The boxes in Fig.8 are the regions concerned in studying the interannual variability of the bloom timing (Figure 12-14).

*254 You should not know about "Photosynthetic activity".*
Reply: We revised "Photosynthetic activity" as "PAR".

*3.3*

*265 Is this about northwestern coast?*

Reply: No. This paragraph is about the climatological distributions of the initiation/termination timings of spring and fall blooms.

*275 Just correlation cannot say "controlled".*

Reply: We have changed "controlled" to "correlated".

*277 favorable?*

Reply: Yes, higher PAR, stronger stratification, and earlier sea ice melting are favorable for earlier spring blooms.

*277-278 I think this sentence mixed spring and fall blooms.*

Reply: Revised.

*280-281 There is no data of increase of PAR.*

Reply: The PAR is shown as shading in Fig.12b.

*282 Is this about deep Japan Basin?*

Reply: Yes. We added "In the deep Japan Basin" in the first sentence of this paragraph.

*288 There is no data of change wind.*

Reply: The wind speed is shown as shading in Fig.13b

*290-291 It is already written.*

Reply: Deleted.

*292 Delete "The correlation show that".*

Reply: Deleted.

*4. Discussion*
*This is very poorly written.*

Reply: We would greatly appreciate if the reviewer could give constructive suggestions and comments.

*5. Summary*

*350 "SSC bloom" should be "phytoplankton bloom".*

Reply: Revised.

*(1) I am not sure which part gives this conclusion. At least, comparison of correlations of different processes is not shown.*

Reply: We are sorry for the misleading expression. This conclusion is revised as follows:

> JES SSCC is characterized by bimodal blooms in spring and fall. Different physical environmental factors are responsible for the JES SSCC evolution during the raise and decline stages of the spring and fall blooms. Increased PAR, weakened winds and melting sea ice are the controlling factors inducing the initiation of spring blooms. Enhanced stratification, which inhibits the upward transport of nutrient-rich waters to the upper layer to compensate for the rapid consumption of nutrients, is the controlling factor inducing the termination of spring blooms. Destratification and active oceanic dynamic processes are favorable factors for fall blooms because they transport nutrient-rich waters to the upper layer. Declining PAR tends to play a negative role and serve as a limiting factor for phytoplankton growth when it is reduced to a certain extent, thereby resulting in the termination of fall blooms.

*354 PAR*

Reply: Revised.

*(3) (4) It is very difficult to understand the idea from Fig. 14 (just a time series plot).*

Reply: The points (3) and (4) are derived from not only Fig. 14, but also the rest figures (e.g., Figure 11, 12 and 13) and logical judgment. All the information mentioned in (3) and (4), e.g., the timings, durations, magnitudes, trends, can be easily identified by these figures.

*(5) "Relative to the AO" As English, it is difficult to understand the contrast to ENSO. I do not understand how ENSO counterbalance in SSC.*

Reply: We are sorry for the misleading expression. ENSO event-induced anomalies somehow result in contradictory SSC responses in the JES. For instance, El Niño events tend to be followed by weak wind speeds that favor early spring blooms (Yamada et al., 2004). On the other hand, El Niño events tend to co-occur with warm winters and thus less sea ice, thus favoring negative spring SSC anomalies in the JES (Hong et al., 2001; Park et al., 2014). In terms of fall blooms, higher SSCs and

longer bloom durations are expected, as active typhoons benefit the vertical transport of nutrient-rich waters during El Niño years (Goh and Chan, 2010). However, at the same time, the warmer background states that occur during the summer-fall seasons in the JES act to inhibit SSC increases and thus cancel out the role of typhoons (Cheon, 2020).

The point (5) is revised as follows:

The interannual variability in JES SSCs are not statistically correlated with ENSO, which tends to induce atmospheric and/or oceanic anomalies that favor a counterbalanced response in SSC.

*Fig. 8 The areas averaged for (c) and (d) are?*

Reply: We revised the original sentence as "The areas averaged SSC for (c) and (d) are marked with black boxes in (a) and (b), respectively. ".

*Fig. 9 Is the value from the box in Fig. 8?*

Reply: No. Y-axis in Figure 9 indicate the corresponding time coefficients of the first EOF mode of SSC anomalies in the JES.

*Fig. 10 Is the value from the box in Fig. 8?*

Reply: No. The boxes in Fig.8 are the regions concerned in studying the interannual variability of the bloom timing (Figure 12-14).